# Identification of Olives Using In-Field Hyperspectral Imaging with Lightweight Models

**DOI:** 10.3390/s24051370

**Published:** 2024-02-20

**Authors:** Samuel Domínguez-Cid, Diego Francisco Larios, Julio Barbancho, Francisco Javier Molina, Javier Antonio Guerra, Carlos León

**Affiliations:** Department of Electronic Technology, Escuela Politecnica Superior, Universidad de Sevilla, 41011 Seville, Spain

**Keywords:** hyperspectral imaging, olives, precision agriculture, machine learning, pattern recognition

## Abstract

During the growing season, olives progress through nine different phenological stages, starting with bud development and ending with senescence. During their lifespan, olives undergo changes in their external color and chemical properties. To tackle these properties, we used hyperspectral imaging during the growing season of the olives. The objective of this study was to develop a lightweight model capable of identifying olives in the hyperspectral images using their spectral information. To achieve this goal, we utilized the hyperspectral imaging of olives while they were still on the tree and conducted this process throughout the entire growing season directly in the field without artificial light sources. The images were taken on-site every week from 9:00 to 11:00 a.m. UTC to avoid light saturation and glitters. The data were analyzed using training and testing classifiers, including Decision Tree, Logistic Regression, Random Forest, and Support Vector Machine on labeled datasets. The Logistic Regression model showed the best balance between classification success rate, size, and inference time, achieving a 98% F1-score with less than 1 KB in parameters. A reduction in size was achieved by analyzing the wavelengths that were critical in the decision making, reducing the dimensionality of the hypercube. So, with this novel model, olives in a hyperspectral image can be identified during the season, providing data to enhance a farmer’s decision-making process through further automatic applications.

## 1. Introduction

Over the past three decades, global olive production has significantly increased from 2.403 million to 5.671 million tons between 1990/1991 and 2020/2021 [1]. This production is divided into table olives and olive oil products, with the majority of these products coming from Mediterranean basin countries such as Spain, Turkey, Greece, or Italy. Olive cultivation is a significant industry in these countries, responsible for producing 32.6%, 15.1%, 14%, and 9.6%, respectively, of the world’s olives [1]. Monitoring the status of olives to ensure optimal nutrition, water, and fertilizer is crucial for producing a high-quality final product. Additionally, experts have traditionally classified olives using the well-known Maturity Index (MI) through physical inspection [2,3,4]. This physical approach requires the destruction of the olives, picking the olives from the tree and cutting them to see their inside. An automatic technique should be developed to eliminate the need for constant monitoring by an expert in the field. Precision agriculture, a technology-driven approach to farming, has been increasingly utilized in recent years to achieve this goal.

Precision agriculture involves managing farm inputs to maximize output. In this regard, the objective of the management policy could be to increase the yield and quality or even to reduce the inputs to maximize profits [5]. Typical farm inputs include fertilizers, irrigation water, herbicides, or plant additives. The plants’ requirements are monitored to determine the precise quantity and type of product to apply [6]. To utilize this approach, it is necessary to invest in sensor networks. Various types of sensors, including water flow sensors, humidity sensors, and soil sensors, must be deployed to measure the inputs [7]. Thus, these sensors must send the information using wireless communication to a platform which will manage the information. This information can be process on the edge (edge computing) or on the cloud (cloud computing). Nevertheless, a wireless sensor network or the Internet of Things (IoT) paradigm is required to acquire the information of the physical systems [8]. After aggregating the information and using support decision algorithms, the next step is to make the necessary adjustments to the system. In this proposal, the Cyber–Physical System (CPS) paradigm plays a major role [9]. A CPS digitalizes the characteristics of a physical entity through the sensor monitoring of its variables. The information is then sent to the digital world and processed using specific models and algorithms. The decision criteria are applied to the control units to modify the entity’s characteristic variables and change its operational state. Therefore, remote sensing is a feasible technique for monitoring fields.

Three sources of information are available in the remote sensing paradigm: satellites, aerial devices such as unmanned aerial vehicles (UAVs), and in-field devices [10]. Remote sensing can be used to manage crops and develop the precision agriculture paradigm. The use of satellites is particularly extensive and has been employed since the launch of Landsat 1 in 1972. The most recent Landsat satellite, Landsat 9, was launched in 2021 [11]. Landsat is a satellite program of NASA, but there are also missions in other countries, such as the Sentinel program in Europe [12]. These satellite missions collect data that are available for the community, such as VIIRS [13], MODIS [14], or HSL [15], which contain a collaboration between Landsat and Sentinel. The monitoring of natural environments and some crops is available through these satellite missions. The first satellite mission had RGB imaging capabilities, but nowadays a wide bandwidth of the spectrum is available for using with these tools. Using these data, several indicators such as NVDI, WSI, and red-edge slope are defined to evaluate and correlate the crops’ properties. This approach is used to obtain general field information, but image resolution is not high enough to identify individual olives. In our case, other approaches with higher resolution must be considered.

The use of UAVs or drones is a common approach in precision agriculture [16]. These platforms improve data resolution and quality [17], and reduce the waiting period for information compared to satellite data [18]. In addition, flights can be organized and programmed to acquire all necessary field details. However, this approach has several issues; for instance, it consumes a lot of power, which calls for the development of low-consumption portable systems. Although the autonomy of UAVs has increased in recent years, it may still be necessary to stop and change the battery of the system when monitoring a crop and acquiring information [19]. Additionally, weather conditions can affect the flight plan, and wind can impact data acquisition. The resolution of the UAV is limited, making it insufficient for certain applications as it is in our case. Monitoring fruit, in particular, can be challenging due to the high resolution required and the obstruction caused by canopies.

A methodology exists for analyzing fields directly on the surface. This is called in-field monitoring, which involves using cameras or hand-held sensors to register the crops [20,21]. The purpose of this approach is to gather the required data by taking the equipment directly to the field of interest. This ensures that resolution is not a problem when selecting which area of the crops to monitor. This technique is particularly useful for applications where it is not possible to detect and obtain information about fruits by flying UAVs over the canopy. However, it is important to note that this technique can be costly due to the equipment required and the need to deploy cameras or sensors in the field.

After monitoring the field and crop, the next step is to identify areas of interest or the focus of the study. Classical image processing techniques can be used to identify regions of interest in images. For example, patterns can be used to identify olive trees in satellite images [10,22,23]. Using several indicators from satellite images, it is possible to provide decision support to farmers. In addition to classical methods, patterns can be searched for in the images. This is particularly useful in specific applications such as fruit tree plantations, where the trees follow a specific pattern. However, in cases where there is no discernible pattern, such as in our case, these methods are not appropriate. In addition to these solutions, it is possible to use artificial intelligence to identify objects in images [24,25]. This approach involves processing the images to find a pattern that has been previously identified in training. For RGB images, there are solutions based on Artificial Neural Networks (ANNs) that are trained with the image and perform several operations to identify patterns of interest. Convolutional Neural Networks (CNNs) are an evolution of Artificial Neural Networks (ANNs) that are specifically designed for image processing [26]. They use a kernel to reduce the dimension of the image and identify patterns. Some state-of-the-art models based on CNNs include Yolo [27,28], ResNet [29], and VGG [30], among others. However, these models are often criticized for their large size and slow processing time [31]. For this purpose, we searched for a model that takes into account both size and execution time. Moreover, images that have a similar background color to the detected object may not provide sufficient information. This is especially problematic when detecting green olives against a green background caused by leaves. To address this chromatic issue, hyperspectral technology is used to gather more spectral information. Multiple wavelengths in the visible and NIR spectra are used to evaluate the quality of the olive, instead of relying solely on the three bands that contain red, blue, and green values. This approach provides additional information [32].

In this type of model, a pretrained model is utilized and then the internal weights of the network are adjusted for the specific application through transfer learning [33]. Additionally, there are transformer-based approaches for remote sensing applications in image classification [34,35], as well as other approaches for regression [36,37], rather than natural language processing. These models are modifications of the attention-based model presented by Vaswani [38]. These models exhibit high performance, but they are computationally intensive. Our objective is to develop a lightweight model capable of identifying olives in images. Since the monitoring will be conducted directly in the field, the deployed processing units must be able to load and execute inferences from the model on the edge. For this purpose, spectral information is utilized along with machine learning models such as Decision Trees, Logistic Regressors, and ANN with a MultiLayer Perceptron. The MultiLayer Perceptron is a classical model with a few hidden layers and neurons [39].

The main contributions of the paper are as follows:(1)A lightweight classifier is proposed for the real-time identification of olives in images taken in olive groves, which is a problem that has not been extensively addressed in the literature.(2)Spectral band analysis is used for wise dimensional reduction. A spectral band analysis was performed to identify the critical bands for the identification problem through wise dimensional reduction using wavelengths in the visible and NIR spectra.(3)Comparison with other state-of-the-art techniques. We compared the state-of-the-art techniques for real-time in-field object identification using a lightweight classifier.

## 2. Materials and Methods

Hyperspectral images of olives on the tree were captured periodically throughout the olive season using a non-invasive, in-field approach. with the Specim IQ model from Specim [40]. This device permits the acquisition of images with 512 × 512 pixels and 204 bands, covering a range of 397 to 1004 nm of the spectrum, with a resolution of 7 nm for each band. Apart from the hyperspectral camera, a 5-megapixel RGB camera is available in the device to acquire the scene information. The images of the olives were acquired at the end of the fruit set phase, and the growth process was recorded until the olives reached the maturity stage, characterized by their outer color turning to purple. The study is focused on green olives, so the monitoring stage concludes with the color change that takes place during the ripening phase. The monitoring period was conducted from May to September, with images taken at a distance of 1 to 2 m from the tree and avoiding direct sunlight. Specifically, they were collected between 9 a.m. and 11 a.m. UTC during the years 2021 and 2022. The olive grove is located in Andalusia, southern Spain, near the city of Seville. Its GPS coordinates are 37.393999, −6.122260.

A white reference tile made of PolyTetraFluoroEthilene (PTFE) was included in all images to standardize illumination conditions. The tile measures 20 cm × 20 cm. PTFE exhibits a uniform response across the electromagnetic spectrum in the 400 to 1000 nm range [41], making it detectable by the camera.

### 2.1. Dataset and Data Management

The dataset consists of hyperspectral images, denoted by $H\in R^{512\times 512\times 204}$, which are referred to as hypercubes in the literature. The hypercube is formed by the two coordinates of an image and a third dimension that has the information of the bands. While an RGB image has 3 bands, hyperspectral images have more bands with reflectance values of the wavelength of the spectral. In this paper, we denote the horizontal coordinate of the image as x, and the vertical coordinate as y, while the wavelength of the image is denoted as z. So, regarding these criteria, we have 3 vectors: $x\in R^{512}$, $y\in R^{512}$, and $z\in R^{204}$. As described before, the images are taken from May to September, creating a database of 400 hyperspectral images during the season of 2022. We evaluated the images at pixel level, to identify if the pixel belongs to an olive or not, only by its spectral signature. In this regard, the models that we discuss in this paper are not aware of the context. Each image contains 204 values per pixel and has a resolution of 512 × 512 pixels, resulting in a total of $53.477\times 10^{6}$ values per hyperspectral image that need to be processed.

The olives in the images were manually classified, and a mask file was generated to indicate the classes. The two classes identified in this novel approach, which was conducted under non-controlled lighting conditions, are ‘olive’ and ‘non-olive’, denoted by 1 and 0, respectively. The resulting mask image, $M\in R^{512\times 512}$, is used as the target class, so, for a specific pixel of the image, we have:$$i,j,k\in N\;and\;i,j\in \left[0,\;511\right]\;and\;k\in [0,\;203]$$
$$H_{i,j}=\left\{z_{0},z_{1},z_{2},\;\ldots ,\;z_{202},z_{203}\right\},\;where\;z_{k}\in R$$
$$M_{i,j}\in \{0,\;1\}$$

The output is an artifact we denoted as $O\in R^{512\times 512}$, which contains elements that belong to two classes resulting from binary classification, $O_{i,j}\in \{0,\;1\}$. Three datasets were created for the algorithm using images from the 2022 season. The training dataset contained approximately 70% of the images, while the validation dataset contained 20% of the total images. The validation dataset images were also taken during the 2022 season, the same season as the training. Finally, the last 10% of the images were from a different season than the training set, which we refer to as the cross-validation dataset. This dataset contained hyperspectral images from the 2021 season, selected randomly to ensure a representative sample. We also made sure to include samples from different days for both training and validation. All images acquired during the 2021 season were used for testing, despite their lower quantity compared to the 2022 season.

### 2.2. Evaluation Metrics

As we sought to develop a classification model, the analysis of model performance would be performed using the metrics extracted from the confusion table. Precision, accuracy, and F1-score [42] were selected as the evaluation metrics to characterize the classification performance. Additionally, we considered size and inference time to determine which model performed better with fewer resources or less time. As is commonly understood, the confusion matrix identifies four categories: True Positive (TP), False Positive (FP), True Negative (TN), and False Negative (FN) [43]. To measure resource usage, we compared model inference time in milliseconds and model size in bytes.

## 3. Framework Presentation

The framework we present consists of three sections: data acquisition and preparation, training loop, and validation and testing. These sections are illustrated in Figure 1, which shows the workflow of information. The arrows indicate the flow of information between points. The dataset is generated offline, and this process is used to train and validate algorithms and models.

### 3.1. Data Acquistion and Preparation

The image was preprocessed to compensate the light conditions and then processed with the algorithm. This process was performed by using the white and black reflectance. These data were available in any image since there was a white PTFE plate present in all images. The compensation expression is the following:$$H_{norm_{i,j,k}}=\frac{H_{i,j,k}-w_{k}\;}{w_{k}-b_{k}}\;\forall \;i,j,k\in N_{0}\wedge k<204\wedge i,j<512$$
where $w,b\in R^{204}$, H is the raw hyperspectral image, $H_{norm}$ is the compensate hyperspectral image, w is the mean reflectance values in the white reference plate, and b represents the mean black values of the camera with the objective closed, acquired before the image acquisition. H and $H_{norm}$ are hypercubes, as described in Section 2.1.

After compensation, the image was stored in a database for the labeling process. The olives in the hyperspectral images were manually identified during this process using image processing software developed by our research team. The software is similar to state-of-the-art solutions for image labeling, such as LabelMe (Release3.0, v5.4.1) for Yolo Images [44]. The software is designed for hyperspectral images following the ENVI standard, while LabelMe is intended for RGB images. It generates a mask file associated with each processed image, which is stored in a database.

### 3.2. Proposed Models and Training

In this section, several models were used to train the images and their associated mask files. The models were evaluated using normalization techniques to tune the training hyperparameters and improve performance based on evaluation metrics. As the model needs to process a large amount of information, we looked for lightweight machine learning-based classifiers. In this regard, models such as Decision Tree (DT), Logistic Regression (LR), and Support Vector Machine (SVM) as a classifier, Random Forest (RF), and clustering models such as k-means or Self-Organizing Maps (SOM) are suitable for our problem. The SVM algorithm used was a C-SVM using a squared L2 penalty and a squared exponential kernel in the isotropic variant using a length score of 0.004902.

These models were trained with the hyperspectral images, extracting the region of interest labelled and reducing the information per image. In this regard, the dataset $S=\left\{X,y\right\}/X\in R^{N\times 204}\wedge y\;\in R^{N}$ where X are the pixels identified as olives or non-olives, and y is the binary target value of the mask, with 1 being an olive and 0 a non-olive. N is the number of pixels that composed the dataset.

As shown in Figure 1, a normalization process was carried out for the model execution. Normalization can be performed in various ways, and the decision of classification can be improved by using normalization techniques [45]. We followed the analysis of the normalization techniques used with hyperspectral images and applied them to our classifiers with the normalized data. The normalization techniques used in this study were Z-norm, Bandnorm, and Max-min. Z-norm involved calculating the mean and standard deviation of the reflectance values across all bands in the image. Bandnorm involved calculating the mean and standard deviation of the values for each band per image. Max-min involved using the maximum and minimum values of the image without normalization. The normalization expressions are as follows:$$X_{zscore_{i,j,k}}=\frac{X_{i,j,k}-\mu }{\sigma }\;\forall \;i,j,k\;\in N_{0}\wedge i,j<512\;\wedge \;k<204$$
$$X_{maxmin_{i,j,k}}=\frac{X_{i,j,k}-minval}{maxval-minval}\;\forall \;i,j,k\;\in N_{0}\wedge i,j<512\;\wedge \;k<204$$
$$X_{bandnorm_{i,j,k}}=\frac{X_{i,j,k}-\mu_{k}}{\sigma_{k}}\;\forall \;i,j,k\;\in N_{0}\wedge i,j<512\;\wedge \;k<204$$
where X is the hypercube of the hyperspectral image, $X\in R^{512\times 512\times 204}$; μ, σ ∈R are the mean value and the standard deviation of the whole hypercube. The variables minval and maxval are the minimum and maximum values of the hypercube, minval∈R : ∄ x in X | x<minval, maxval∈R : ∄ x in X | x>maxval. Then, $\mu ,\;\sigma \in R^{204}$ are the mean and standard deviation of the image for each of the bands of the image.

Several iterations were performed using different models, tuning their hyperparameters to achieve the best results. Each of these well-known models has its own advantages in terms of size, inference time, and classification performance. The criteria we established for selecting the model was the one that achieved high classification performance with the lowest cost. We defined cost as the time and size required to execute the model. The cost associated with the model increases as its size or execution time increases. We defined three classes in the image: olives, non-olive, and white reference. The white reference class is present in all images and has a known signature pattern, which reduces variability with the other classes. The decision is made for each pixel of the image. Therefore, each of the 512 × 512 pixels is evaluated with the model and assigned to one of three classes: olive, non-olive, or white.

### 3.3. Test and Validation

After training and saving the model, it was evaluated using validation images that were not seen during training. The evaluation metrics were calculated for each model and compared to determine their performance. The validation dataset consisted of hyperspectral images from several days of the same season as the model’s training. Additionally, to assess the models’ performance across multiple seasons, they were tested using hyperspectral images from a different season. This is referred to as cross-validation data. The results, including evaluation metrics for both the validation and test datasets, are presented in the results section.

## 4. Results and Discussion

### 4.1. Model Results

In this section, we present the results of the models comparing the classifiers results with the data normalized using the techniques described in Section 3.2. Table 1 contains information about the evaluation metrics of precision accuracy and F1-score. The F1-score is the best metric for comparing the models and selecting the best performance. Accuracy is a simple metric that can cause problems with classification in situations of imbalanced data. Precision only uses the ratio of true cases and is very sensitive to data distribution. Instead, the F1 metric should be used as it provides information on the functionality of the classifier and is less sensitive to imbalanced data than accuracy and precision [42]. For this study, three classes were defined for the classification: white reference, olive, and non-olive. It is worth noting that the number of non-olive pixels is significantly higher than that of olive and white reference pixels in every image. Additionally, we compare the data from one season to another season dataset. Table 1 includes information on the cross-validation dataset, labeled as the ‘test’ column.

As it can be seen, the results of the models are very promising, with results in the classification higher than the 95% in F1-score. This metrics were calculated from a multiclass confusion matrix, averaging the precision in the classification of the three classes. The white reference class validated the image, indicating that it was well taken. The non-olive and white reference class pixels were merged into a single class called non-olive.

Two examples of the confusion matrix of Table 1 were obtained and can be seen in Figure 2. Figure 2a stands for the results of the LR classifier using the Bandnorm technique in the image, while Figure 2b displays the results of the LR classifier without using normalization. Figure 2c,d show the confusion matrix of the SVM with and without normalization, respectively. Furthermore, Figure 3 presents the results obtained from the test dataset.

The data suggest that adjusting image values after white compensation did not significantly enhance classification. The Z-score evaluates the variation of the signatures around a mean and the variation of the values of the images. Then, the bandnorm technique normalizes over each of the bands, so the information of different values of illumination produces major changes which causes an increase in the detection failures. Therefore, illumination conditions have a great impact on this kind of normalization technique. The normalization of the bands affects the values of each band, which can have a negative impact on images with varying illumination conditions. Depending on the dataset, the normalization technique should be evaluated using hyperspectral images [45]. 

After analyzing the classification performance, we conducted an analysis of the cost of the classifier. To do this, we evaluated each classifier during the inference of the validation dataset on the same machine and with the same input. We compared the performance of the classifiers and evaluated their relative time with respect to the lowest value. Table 2 displays the results, indicating that the LR classifier is the lightest and fastest. The system used to perform the inference was a personal computer with an Intel Core i5-8265U CPU with four cores and eight threads working at 1.60 GHz with 8 GB of Ram and an integrated graphical unit.

The results indicate that the SVM classifier is the slowest and heaviest model, while the RF classifier is lighter and faster than the SVM but not as fast as the DT and LR classifiers. In terms of classification performance, the RF model is the worst, while the DT and LR models are both better suited for olive identification due to their low resource usage and high classification performance. The DT classifier is the fastest, although its F1-score is lower than that of the SVM and LR models. The LR model, despite being 44 times smaller than the DT model, has a 31% lower inference time. However, it shows a better F1-score with almost a 4% improvement compared to the DT model. Therefore, for this purpose, the LR model is the best option to use. It is a lightweight model, only 5.5 KB, with an inference time of 231 ms for each hyperspectral image, and a 99% classification F1-score. As an example of this model’s inference, Figure 4a,b display the RGB image converted from a hyperspectral image. The model classifies into three distinct categories: olives, non-olives, and white reference. The non-olives and white reference categories are combined and colored red for visualization purposes, while the olive category is colored green. Figure 4b shows the result of overlaying this mask image onto Figure 4a. The model was tested using data from different seasons. Open data on table olives from the 2021 season are available on the internet [46]. The model was also evaluated to determine if olives from different seasons can be identified. The results of the model can be seen in Figure 5a,b. Furthermore, Figure 6 shows several images of the validation and test datasets overlapped with the classifier’s segmentation.

### 4.2. Critical Band Evaluation

Although the model is small, working with hyperspectral images requires high computing capabilities and hyperspectral sensors. To reduce computing requirements and the number of bands used, an analysis of the most significant bands was carried out, and the contribution of each band was measured for the LR classifier without normalization. This process involves calculating the difference between a baseline metric and a permutation of features. The metrics were calculated by permuting a column of features from the dataset 1000 times, with each feature being changed. Figure 7 depicts the contribution of the T1 significant bands for the LR model. For visualization purposes, only one third of all bands, ranked in order of importance to the classification, were plotted. These values were randomly selected to determine if there are significant differences between bands.

Figure 7 shows that only a few bands contribute significantly to the normalization process. Therefore, we can infer the information using only the upper bands without significantly decreasing the prediction performance. This reduces the model size and the required data for information inference. To test the hypothesis, the models underwent training with a reduced feature set. The original 204 bands were reduced to one third, one quarter, one tenth, and the fifth percentile of the most significant bands. The evaluation metrics and cost were calculated for the models using 33%, 25%, 10%, and 5% of the most significant bands, denoted as P33, P25, P10, and P5 in this paper. The model used for this experiment was the LR model without normalizing the data, as depicted in Table 3. Table 3 displays the results, indicating that the F1-score increases with the number of bands. The SVM model had the least reduction in F1-score, but also the longest inference time, as previously discussed. On the other hand, the LR model remains the smallest and has the shortest inference time when only the top 5% most significant bands are used as input. These bands are 204, 106, 105, 108, 104, 107, 109, 103, 192, and 102, which, respectively, stand for the following wavelengths in nm: 1003.58, 705.57, 702.58, 711.56, 699.60, 708.57, 714.55, 696.61, 966.55, and 693.62.

In this regard, it is possible to reduce the model size to 1 KB and achieve a 98% F1-score for olive detection using 10 bands. The same images used in Figure 6 were also tested with the P5 LR model to observe the impact of a 1% reduction in F-score, as shown in Figure 8. Furthermore, precision agriculture approaches utilize visible images to detect objects such as tomatoes, apples, and strawberries. A visible image is a three-dimensional artifact that represents the color space. By using visible and NIR wavelengths, we can obtain more information about the objects present in the images. According to the analysis carried out in Table 3, these methods should be efficient in terms of time and size. However, as shown in Table 4, a comparison with state-of-the-art techniques was conducted. This table compares the approach of other researchers to acquiring in-field images using deep learning techniques. As far as we know, there are no studies on the identification of olives directly on the field, so other crops were used to compare our model. The aim of our research is different as we focus on real-time applications and use machine learning models to reduce size and computing time. In our case, identification is performed at the pixel level, while other researchers create sub-images using CNNs to classify the presence of olives.

Table 4 demonstrates that our model has been reduced in size, processing time, and F1-score. Although these results are promising, this novel approach should be tested under different conditions and seasons, with varying light conditions. To improve the robustness of this novel model, it is necessary to increase the variability of images under different light conditions, as a reduction in light conditions can increase the probability of misclassification. As exposure time decreases, the measurement range also decreases. Objects in the background may not provide accurate information due to the shorter exposure time.

## 5. Conclusions

In this work, a framework for identifying olives in hyperspectral images was presented. The hyperspectral images were acquired directly in the field without controlled lighting conditions. The reflectance values were compensated using a white PTFE reference, and the image was normalized before the inference of the ML model. The model generated a binary classification output at the pixel level, which was used to segment the image into olive and non-olive categories. A comparison of the DT, LR, SVM, and RF models was conducted to evaluate their performance using normalization techniques. Precision, accuracy, and F1-score metrics were compared. The models achieved a success rate of around 98% in classifying the presence of olives. However, it was observed that the way the image was taken was crucial, as several images identified leaves as olives. The analysis focused on the most important bands in the hyperspectral images. It was found that by using only the top 5% of significant bands, the LR size was reduced by 80% and the inference time was reduced by 74%. However, the model showed a 1% reduction in F1, so it is important to evaluate this F1 reduction for real-time applications.

### Future Work

Subsequent studies should focus on comparing olive identification using only RGB information. Additionally, the dataset should be expanded to include different light conditions and multiple years, creating a comprehensive database of olives during their respective seasons. State-of-the-art complex models can be analyzed to determine the most effective method for detecting olives directly in the field. The dataset can be expanded to include more seasons and different varieties, such as Manzanilla, Hojiblanca, or Arbequina. Other potential applications include evaluating olive maturity, determining the number and size of olives in the image, or implementing global parameters to assist farmers in decision-making.

## Figures and Tables

**Figure 1 sensors-24-01370-f001:**
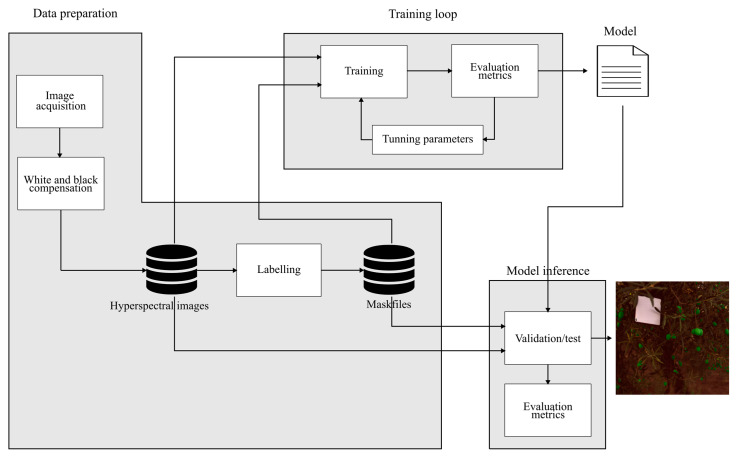
Framework of the proposed model for the identification of the Gordal green olive variety in hyperspectral images.

**Figure 2 sensors-24-01370-f002:**
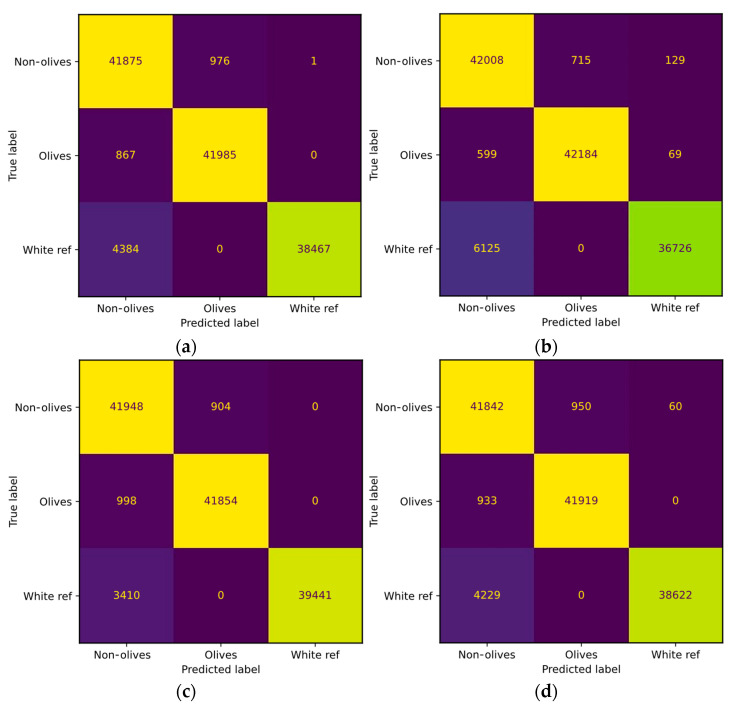
Confusion matrix with the validation dataset. (**a**) LR classifier with the Bandnorm technique. (**b**) LR model without normalization. (**c**) SVM model with the Bandnorm technique. (**d**) SVM model without normalization.

**Figure 3 sensors-24-01370-f003:**
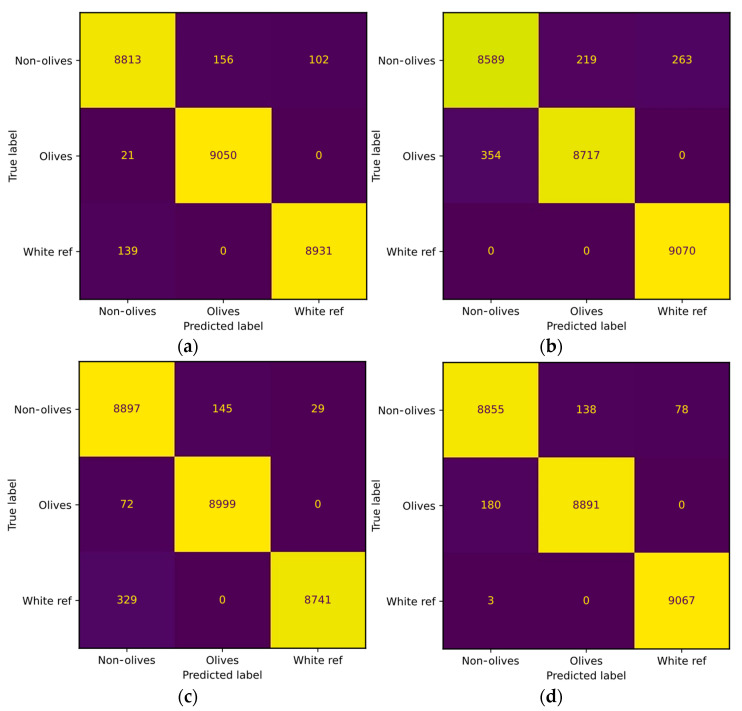
Confusion matrix with the test dataset. (**a**) LR classifier with the Bandnorm technique. (**b**) LR model without normalization. (**c**) SVM model with the Bandnorm technique. (**d**) SVM model without normalization.

**Figure 4 sensors-24-01370-f004:**
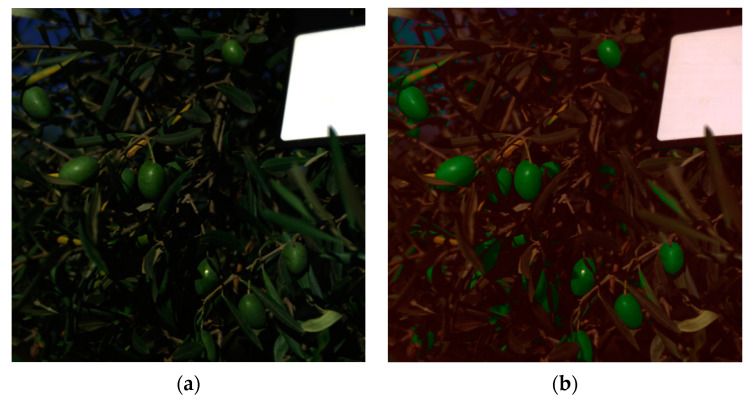
Images of the hyperspectral image converted to RGB (**a**) and the overlapped image with the model results (**b**) of the validation dataset.

**Figure 5 sensors-24-01370-f005:**
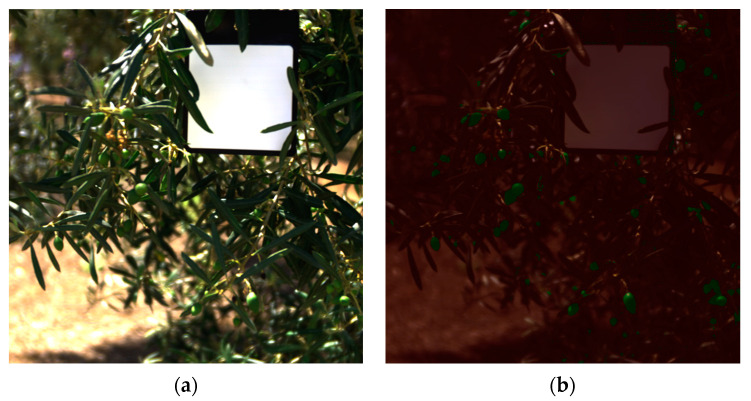
Images of the hyperspectral image converted to RGB (**a**) and the overlapped image with the model results (**b**) of the test dataset, cross-season.

**Figure 6 sensors-24-01370-f006:**
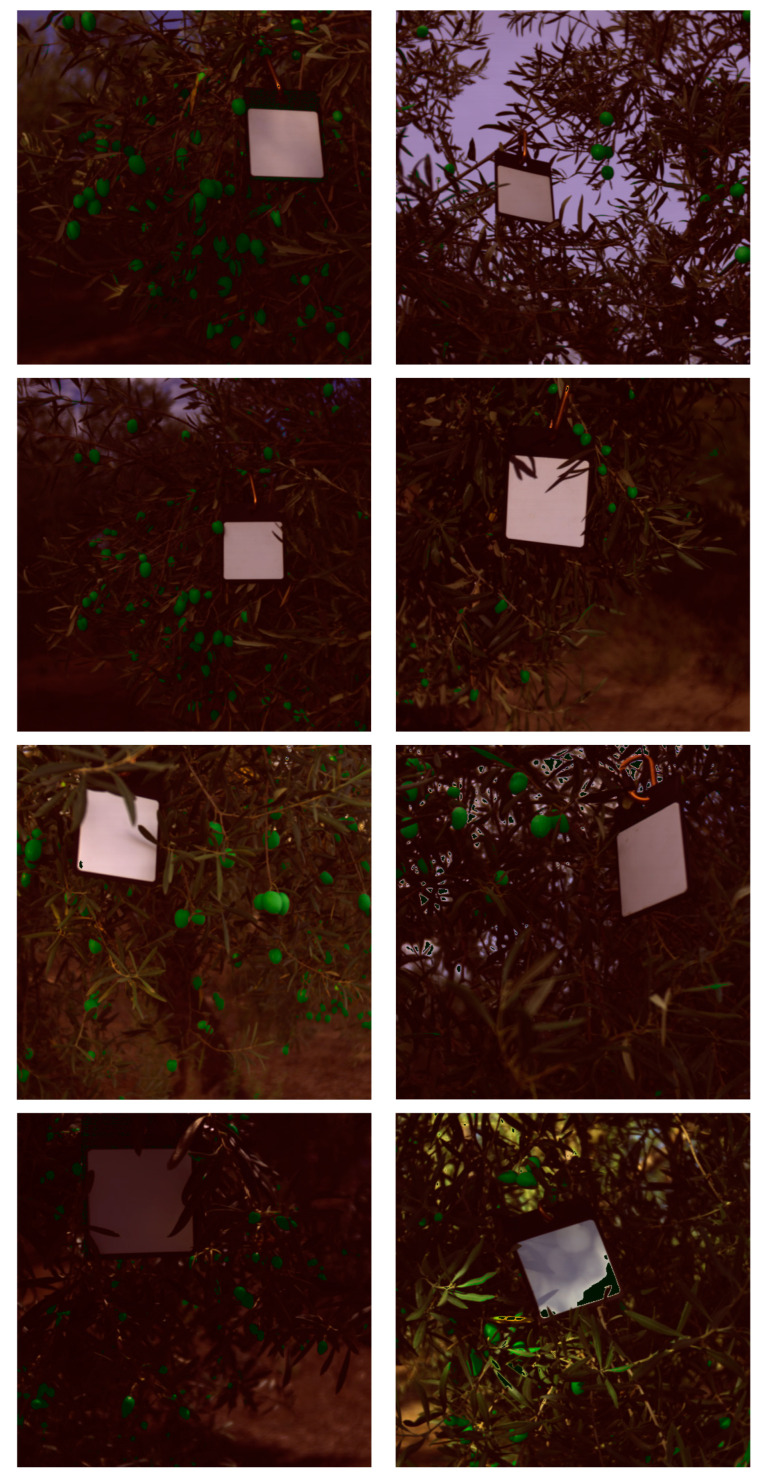
LR classifier output segmentation overlapped with the images of the validation and test dataset.

**Figure 7 sensors-24-01370-f007:**
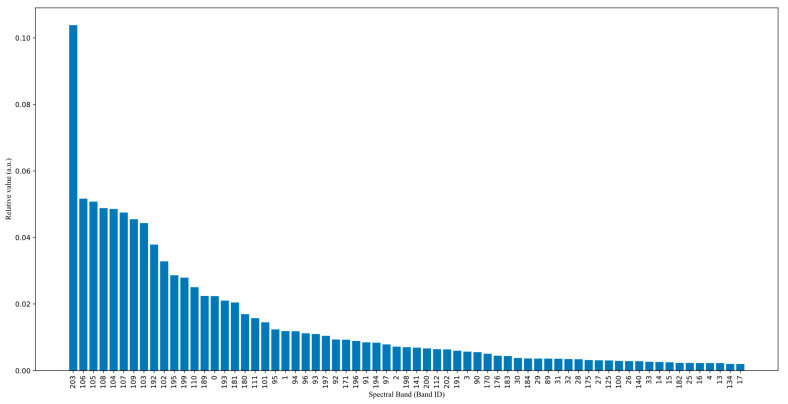
Importance of the 33.33% most significant bands using the LR model without normalization.

**Figure 8 sensors-24-01370-f008:**
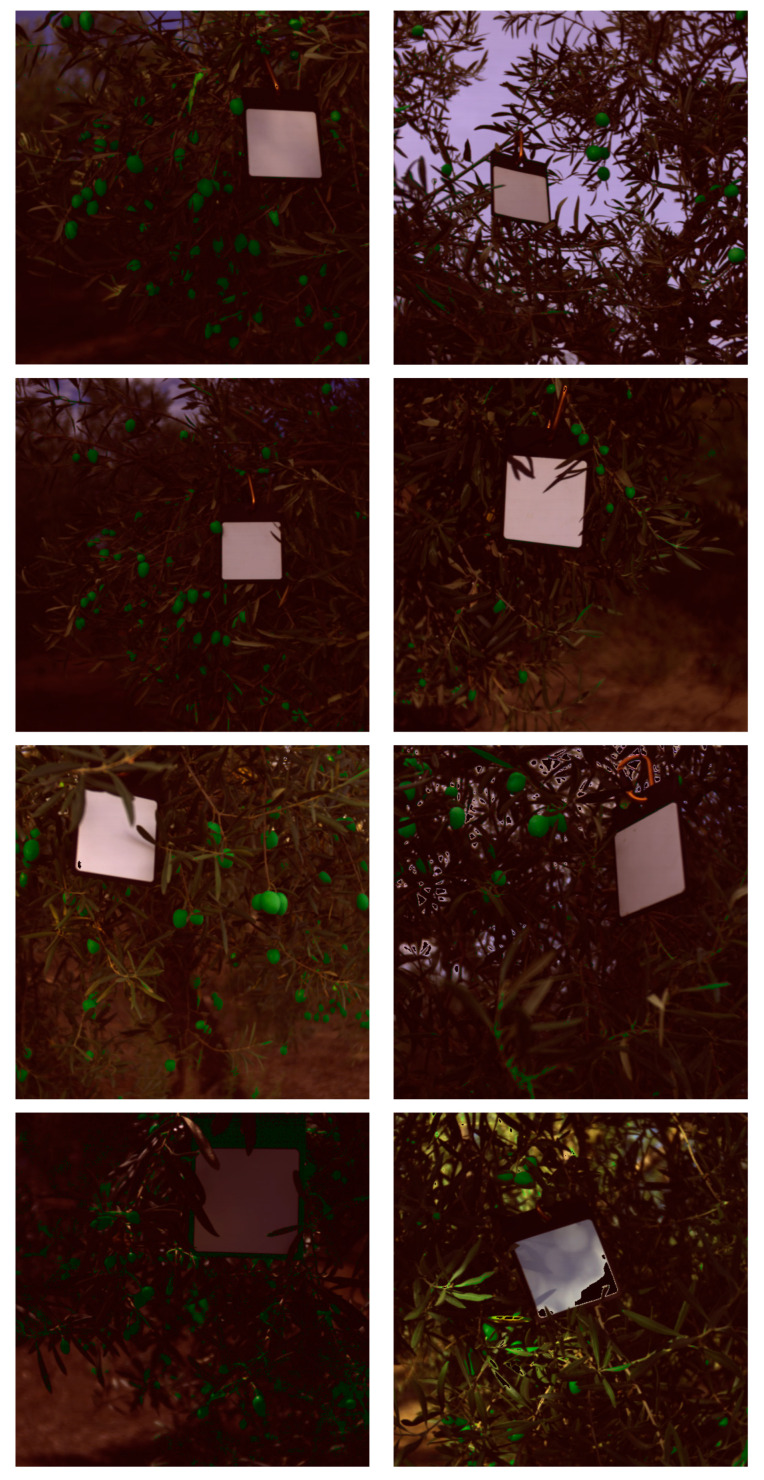
LR classifier output segmentation overlapped with the images of the validation and test dataset using 5% of the spectral bands.

**Table 1 sensors-24-01370-t001:** Evaluation metrics of the validation and test metrics of the models using normalization techniques with the validation and test dataset.

Normalization	Classifier	Precision (%)		Accuracy (%)		F1-Score (%)	
		Validation	Test	Validation	Test	Validation	Test
Bandnorm	LR	98.598	98.988	98.791	99.232	98.641	99.139
	DT	95.777	95.381	96.209	95.142	95.731	94.431
	SVM	98.361	99.008	98.521	99.203	98.335	99.105
	RF	93.481	85.647	94.500	87.569	93.872	86.606
Maxmin	LR	99.225	99.056	99.188	98.835	99.084	98.680
	DT	97.065	96.460	96.613	95.175	96.120	94.367
	SVM	98.967	98.181	99.004	98.218	98.877	97.987
	RF	94.252	96.306	93.772	95.090	92.793	94.273
Z-score	LR	98.865	87.855	99.006	89.534	98.882	88.867
	DT	97.362	93.765	96.855	93.786	96.396	92.882
	SVM	98.720	95.197	98.861	96.068	98.719	95.621
	RF	97.159	92.495	96.246	93.025	95.660	92.081
No normalization	LR	98.949	98.298	99.112	98.501	99.002	98.314
	DT	97.605	96.182	97.777	95.792	97.493	95.173
	SVM	98.348	98.741	98.535	98.831	98.352	98.684
	RF	93.524	85.916	94.518	87.759	93.887	86.889

**Table 2 sensors-24-01370-t002:** Comparison of the evaluation metrics of the models without the normalization technique.

Classifier	F1-Score (%)	Inference Time (ms)	Size (KB)
DT	95.17	164.63	245.97
SVM	98.35	720,248.31	20,841.96
LR	99.00	231.91	5.51
RF	93.89	4855.78	739.96

**Table 3 sensors-24-01370-t003:** Comparison of the evaluation metrics with the different models reducing the feature set of the data without normalizing.

Percentile	Classifier	F1-Score (%)	Inference Time (ms)	Size (KB)
P33	DT	97.094	87.063	270
	LR	98.839	109.156	2.29
	SVM	98.462	470,580.587	6500
	RF	94.287	4854.568	756
P25	DT	97.423	89.960	281
	LR	98.641	94.093	1.92
	SVM	98.330	446,215.157	5110
	RF	93.960	4711.796	768
P10	DT	96.193	81.218	367
	LR	98.079	83.616	1.19
	SVM	98.106	515,363.557	2840
	RF	92.933	5137.557	770
P5	DT	94.375	85.020	462
	LR	98.034	59.558	0.982
	SVM	98.188	762,950.853	2460
	RF	91.202	5259.811	766

**Table 4 sensors-24-01370-t004:** Comparison of the evaluation metrics of the models with state-of-the-art models.

Object Detection	F1-Score (%)	Parameters
InceptionV3 *	83.89	23.90 M
Inception-ResNetV2 *	84.01	55.90 M
Our model	98.03	30.00

* Reference of these results: [24].

## Data Availability

Data are contained within the article.

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
