# Peer review of "Identification of Olives Using In-Field Hyperspectral Imaging with Lightweight Models"

_sensors, 2024, doi:10.3390/s24051370_

Round 1

Reviewer 1 Report

Comments and Suggestions for Authors

1. I think the sentences from lines 135-139 are not necessary. It should be removed from the manuscript.

2. The olive grove is located in southern Spain, near the city of Seville in the region of Andalucia. Please provide the GPS data for this location.

3. Three datasets were created with images for the algorithm, first a training dataset, containing around 70% of the images of the 2022 season, then the 20% of the total images were saved for the validation dataset. Please provide in detail how this selection was done. Random or using a specific algorithm to separate the samples into two datasets.

4.In this case he precision, accuracy and F1-score were the selected evaluation metrics for characterizing the classification performance. Please correct the not he.

5. In this case he precision, accuracy and F1-score were the selected evaluation metrics for characterizing the classification performance. Please show the formula to calculate these figures of merit or provide a reference for this.

6. Please use a consistent term. For example: no-olive and non-olive. The two terms are different in meaning, I think. In binary modeling, it is common to use X and non-X, for example.

7. From Figure 7, please show the criteria or threshold values (Relative value) for the band selection. Peak selection is possible? 

8. Please provide detailed information on the SVM algorithm: C-SVMC or nu-SVMC. What is the kernel function used in the SVM?

Author Response

Dear Dr./ Mr./Ms.,

Thank you for giving me the opportunity to submit a revised draft of my manuscript titled “Identification of olives using in-field hyperspectral imaging with lightweight models” to Sensors.  We appreciate the time and effort that you and the reviewers have dedicated to providing your valuable feedback on my manuscript. We are grateful to the reviewers for their insightful comments on our paper. We have been able to incorporate changes to reflect most of the suggestions provided by the reviewers. We have highlighted the changes within the manuscript.

Here is a point-by-point response to the reviewers’ comments and concerns.

Comment 1:     I think the sentences from lines 135-139 are not necessary. It should be removed from the manuscript.

Response: Thank you for pointing this out. We agree with this comment. Therefore, we have removed these lines that does not add value to the paper and are just descriptive of the organization. Lines have been removed.

Comment 2: The olive grove is located in southern Spain, near the city of Seville in the region of Andalucia. Please provide the GPS data for this location.

Response: We agree with this and have incorporated your suggestion throughout the manuscript. On line 166, the GPS coordinates have been added to the olive grove location.

Comment 3: Three datasets were created with images for the algorithm, first a training dataset, containing around 70% of the images of the 2022 season, then the 20% of the total images were saved for the validation dataset. Please provide in detail how this selection was done. Random or using a specific algorithm to separate the samples into two datasets.

Response: Thank you for the suggestion, it seems it was not clear in the text. For this reason, lines 206 to 209 have been modified. The selection of the images was random, although the number of images per days was selected to have the maximum amount as possible, the same number of images was used.

Comment 4: In this case he precision, accuracy and F1-score were the selected evaluation metrics for characterizing the classification performance. Please correct the not he.

Response: Thank you, line 212 was modified, the misspelling was corrected for better understanding as you pointed out.

Comment 5: In this case he precision, accuracy and F1-score were the selected evaluation metrics for characterizing the classification performance. Please show the formula to calculate these figures of merit or provide a reference for this.

Response: We have, accordingly, modified line 213 and a reference was added to recall the calculation of state-of-the-art metrics as you suggested.

Comment 6: Please use a consistent term. For example: no-olive and non-olive. The two terms are different in meaning, I think. In binary modeling, it is common to use X and non-X, for example.

Response: Thank you for the suggestion, we have changed the term “no olive” to “non-olive” throughout the manuscript.

Comment 7: From Figure 7, please show the criteria or threshold values (Relative value) for the band selection. Peak selection is possible?

Response: Agree. We have revised the document to emphasize this point. The number of bands criterion was arbitrarily selected, the number of bands were reduced to see the performance of the model with lesser information. Each of the bands was permutated with the classifier comparing the output at a baseline with all the band. The importance is not a threshold score to include the band but the performance of the model with less information. From lines 408 to 411 a text was added to explain this a little more as it was not clear.

Comment 8: Please provide detailed information on the SVM algorithm: C-SVMC or nu-SVMC. What is the kernel function used in the SVM?

Response: Thank you for pointing this out. We have added lines 256 to 257 explaining in detail the SVM algorithm used as was not explained before.  We used a C-SVM classifier using a rbf kernel. It was used a L2 penalty and length score of 0.004902 in the kernel.

Reviewer 2 Report

Comments and Suggestions for Authors

In this manuscript, the authors described “Identification of olives using in-field hyperspectral imaging with lightweight models”. However, there are some problems in the study, here are some points of improvement:

Abstract:

Need to add research findings to the article.

Introduction:

1. The detailed background on olives is commendable, but after establishing the importance of detection, a concise review of early methods used for this purpose would be enriching. Authors should thoroughly describe traditional techniques relevant to determination and point out the limitations of these techniques to better highlight the advantages of hyperspectral imaging technology.

2. The content from lines 135 to 139 does not require narration and can be deleted. The main research focuses and innovative points need to be added.

Materials and Methods:

1. The authors mention in the abstract that the data were collected from 9:00-11:00am, and then in Materials and Methods they say it was from 10a.m to 12p.m (line 152), is this a misrepresentation?

Results and discussion:

1. It is suggested that the authors combine Table 1 and Table 2 into one table so that the modeling set and test set can be compared better.

2. The article lacks a discussion on the model performance metrics obtained under different normalization methods. According to the results in Table 1 and Table 2, the F1-score of the model without normalization is higher than that of most models with normalization. Why is this the case?

3. The number of decimal places of the table data in the article should be consistent.

4. Please explain why the F1-score is the best classification metric for the classification model of this dataset?

5. In 4.2 Critical Bands evaluation, could you provide a detailed explanation of how these 33.33% most significant bands were determined, specify the range of wavelengths included in the selected 5% spectral bands, and explain what information about olives is reflected by these 5% of the spectrum?

6. In 4.2 Critical Bands evaluation, can better results be achieved by using a variety of wavelength selection algorithms to choose the optimal wavelengths?

7. In the results shown in Fig. 8, a large area near the whiteboard in the first photo of the fourth row is mistakenly marked as green. The lighting in this photo seems weaker than in the other photos. Could the weaker lighting increase the probability of misclassification? The authors should analyze the impact of different lighting conditions on the classification performance of the LR model.

Comments on the Quality of English Language

Minor editing of English language required

Author Response

Dear Dr./ Mr./Ms.,

Thank you for giving me the opportunity to submit a revised draft of my manuscript titled “Identification of olives using in-field hyperspectral imaging with lightweight models” to Sensors.  We appreciate the time and effort that you and the reviewers have dedicated to providing your valuable feedback on my manuscript. We are grateful to the reviewers for their insightful comments on our paper. We have been able to incorporate changes to reflect most of the suggestions provided by the reviewers. We have highlighted the changes within the manuscript.

Here is a point-by-point response to the reviewers’ comments and concerns.

ABSTRACT

Comment 1: Need to add research findings to the article.

Response: Thank you for pointing this out. We have added finding of the research were added to the abstract, lines 17 to 20 were added.

INTRODUCTION

Comment 1: The detailed background on olives is commendable, but after establishing the importance of detection, a concise review of early methods used for this purpose would be enriching. Authors should thoroughly describe traditional techniques relevant to determination and point out the limitations of these techniques to better highlight the advantages of hyperspectral imaging technology.

Response: We have, accordingly, revised the introduction where different methods to classify are explained. However, to emphasize the limitation of RGB images, lines 119 to 127 were added to clarify this topic. Problems with the RGB are specified and referenced and the information that provide the HSI is showed. 

Comment 2: The content from lines 135 to 139 does not require narration and can be deleted. The main research focuses and innovative points need to be added.

Response: These lines were removed as suggested, these lines were not adding any value to the text. The main focus and innovative points are explained during the introduction summarize in lines 141 to 151.

MATERIALS AND METHODS

Comment 1: The authors mention in the abstract that the data were collected from 9:00-11:00am, and then in Materials and Methods they say it was from 10a.m to 12p.m (line 152), is this a misrepresentation?

Response: Thank you, as you suggest this was a misrepresentation, it was a confusion between local and UTC time. Each time was changed and specify to be in UTC time.

RESULTS AND DISCUSSION

Comment 1: It is suggested that the authors combine Table 1 and Table 2 into one table so that the modeling set and test set can be compared better.

Response: Agree. We have, accordingly, merged Table 1 and Table 2 into Table 1 as suggested to compared better the results.

Comment 2: The article lacks a discussion on the model performance metrics obtained under different normalization methods. According to the results in Table 1 and Table 2, the F1-score of the model without normalization is higher than that of most models with normalization. Why is this the case?

Response: Thank you for the comment, we have changed lines 329 to 336 discuss the different normalization methods. Performing a normalization technique, modify the information of the reflectance sensor, changing the values modify the variability of the images. Z-score modify in the range and only evaluate the variation from an average point and its variation of the image, then, bandnorm technique normalize over each of the bands available of data, so the information for different values of illumination changes a lot which provokes an increase in the detection failures.

Comment 3: The number of decimal places of the table data in the article should be consistent.

Response: Agree. We have revised the number of decimals throughout the manuscript to make it consistent.

Comment 4: Please explain why the F1-score is the best classification metric for the classification model of this dataset?

Response: Agree. The F1-score is used as evaluate the performance of the classifier and it is sensitive to data distribution, regarding imbalanced data. Accuracy is the simplest way to evaluate the functionality of a classifier, but it is highly dependent on the data distribution. Precision only looks to the true positive, so only on the positive case, those cases where the classifier is missed on classifying the negative value is not measured. F1-score combines recall and precision and gives information of the functionality of the classifier in both situations, on the positive and negative case. To clarify why the F1-score is used, lines 313 to 318 of the paper have been modified. Reference [42], “Learning from Imbalanced Data”, was added explaining this problem.

Comment 5: In 4.2 Critical Bands evaluation, could you provide a detailed explanation of how these 33.33% most significant bands were determined, specify the range of wavelengths included in the selected 5% spectral bands, and explain what information about olives is reflected by these 5% of the spectrum?

Response: Thank you for pointing this out. The selection of 33% was randomly used and figure 7 is for visualization purpose to see how each of the bands affect to the decision-making of the models. As there are some bands that have more importance than others. The bands importance was sorted and then the quantity of the bands was reduced to assess if the model was able to continue classifying with reduced information.

The range of wavelength that contains in the 5% are 0.05*204 ≈ 10 bands which are: 204, 106, 105, 108, 104, 107, 109, 103, 192, 102, which stands for the following wavelengths in nanometers respectively: 1003.58, 705.57, 702.58, 711.56, 699.60, 708.57, 714.55, 696.61, 966.55, 693.62.

In these bands they have importance in the visible and several of NIR bands. Visible wavelength range goes from 380 to 700 nm and NIR goes from 700 to 2500nm the NIR spectra. The visible components, contains the values of red component which contain information of how green or purple the olive is which is not close to the color of the bands. In the manuscript, Section 4.3 was changed adding the explanation that was missing and specifying the range of wavelengths.

Comment 6: In 4.2 Critical Bands evaluation, can better results be achieved by using a variety of wavelength selection algorithms to choose the optimal wavelengths?

Response: Thank you for this suggestion. It would have been interesting to explore this aspect. However, in the case of our study, it seems slightly out of scope because we wanted to focus on the detection of the olives rather using several algorithms to evaluate the importance of the bands. We know that there are several algorithms to get the significant information but, in our work, we didn’t compare them. In the evaluation there were some of the bands that do not contribute positively to the classification, they introduced noise to the model. In this regard, evaluating the score of each of the bands can be used to assess the importance of the bands and discard those whose information is not improving the classification. Nevertheless, we are really thankful for this comment, and we are thinking about explore this line in future works.

Comment 7: In the results shown in Fig. 8, a large area near the whiteboard in the first photo of the fourth row is mistakenly marked as green. The lighting in this photo seems weaker than in the other photos. Could the weaker lighting increase the probability of misclassification? The authors should analyze the impact of different lighting conditions on the classification performance of the LR model.

Response: You have raised an important point here. We have tested the model changing the values of lighting conditions, reducing lightning the probability of misclassification increased. Lines 457 to 4461 explain this effect. In order to solve this problem with low lighting area we are willing to develop in future work dynamic lighting algorithms for this case. The darker the image is the lesser information of the object is saved in the image. The hyperspectral image acquires the information using the light so, as the exposure time is reduced the images gets darker reducing the information that contains.

Round 2

Reviewer 2 Report

Comments and Suggestions for Authors

The table 2 has been deleted in the article, thus, an update to the table is needed.

Comments on the Quality of English Language

Minor editing of English language required

Author Response

Dear Dr./ Mr./Ms.,

Thank you for giving me the opportunity to submit a second revised draft of my manuscript titled “Identification of olives using in-field hyperspectral imaging with lightweight models” to Sensors.  We appreciate the time and effort that you and the reviewers have dedicated to providing your valuable feedback on my manuscript to solve the final details. We are grateful to the reviewers for their insightful comments on our paper. We have been able to incorporate changes to reflect most of the suggestions provided by the reviewers. We have highlighted the changes within the manuscript.

Here is a point-by-point response to the reviewers’ comments and concerns.

Suggestion 1: The table 2 has been deleted in the article, thus, an update to the table is needed.

Response: Thank you for pointing this out. We have revised and changed the indexes of all tables and images to make them consistent.  

Suggestion 2: Minor editing of English language required.

Response: We have revised the whole manuscript to improve English, checking grammar, verb tenses and correcting misspellings. 
